# CX3CL1 (Fractalkine)-CX3CR1 Axis in Inflammation-Induced Angiogenesis and Tumorigenesis

**DOI:** 10.3390/ijms25094679

**Published:** 2024-04-25

**Authors:** Dariusz Szukiewicz

**Affiliations:** Department of Biophysics, Physiology & Pathophysiology, Faculty of Health Sciences, Medical University of Warsaw, 02-004 Warsaw, Poland; dariusz.szukiewicz@wum.edu.pl

**Keywords:** chemokine CX3CL1, fractalkine, CX3CR1, fractalkine receptor, CX3CL1/CX3CR1 axis, angiogenesis, tumorigenesis, inflammation, inflammation-induced angiogenesis

## Abstract

The chemotactic cytokine fractalkine (FKN, chemokine CX3CL1) has unique properties resulting from the combination of chemoattractants and adhesion molecules. The soluble form (sFKN) has chemotactic properties and strongly attracts T cells and monocytes. The membrane-bound form (mFKN) facilitates diapedesis and is responsible for cell-to-cell adhesion, especially by promoting the strong adhesion of leukocytes (monocytes) to activated endothelial cells with the subsequent formation of an extracellular matrix and angiogenesis. FKN signaling occurs via CX3CR1, which is the only known member of the CX3C chemokine receptor subfamily. Signaling within the FKN-CX3CR1 axis plays an important role in many processes related to inflammation and the immune response, which often occur simultaneously and overlap. FKN is strongly upregulated by hypoxia and/or inflammation-induced inflammatory cytokine release, and it may act locally as a key angiogenic factor in the highly hypoxic tumor microenvironment. The importance of the FKN/CX3CR1 signaling pathway in tumorigenesis and cancer metastasis results from its influence on cell adhesion, apoptosis, and cell migration. This review presents the role of the FKN signaling pathway in the context of angiogenesis in inflammation and cancer. The mechanisms determining the pro- or anti-tumor effects are presented, which are the cause of the seemingly contradictory results that create confusion regarding the therapeutic goals.

## 1. Introduction

### 1.1. Angiogenesis

A well-developed microcirculatory network ensuring optimal blood–target tissue interactions is essential for preserving optimal metabolism. No metabolically active tissue in the body is more than a few hundred micrometers from a blood capillary, which is formed via the process of angiogenesis [1]. Angiogenesis involves two different mechanisms of vessel formation and two types of vessels being formed. Angiogenesis, in a strict sense, describes the formation of new blood vessels from pre-existing functional vessels via sprouting or splitting (also known as intussusceptive angiogenesis). This process is accompanied by the migration and proliferation of endothelial cells (ECs) [2]. However, de novo vessel formation may occur as a result of the differentiation of endothelial progenitor cells (EPCs) and their subsequent integration into the vascular wall. Human EPCs are generally defined as circulating cells expressing a variety of markers on their surface that are similar to the markers expressed by vascular ECs, adhere to the endothelium at sites of hypoxia/ischemia or vascular injury, and participate in neoangiogenesis, including tumor angiogenesis [3,4]. Many studies revealed heterogeneity in blood ECs, where a certain pool of cells expresses endothelial and hematopoietic antigens, and others express mature or immature endothelial markers [5]. Some studies indicate that the direct source of EPCs is not the bone marrow, as was commonly assumed, but the activation of cells with EPC properties residing in tissues (e.g., heart muscle) [6,7]. In contrast to sprouting angiogenesis, splitting angiogenesis is a rapid recovery adaptation of the existing microvascular network. However, this process only relies on the proliferation of ECs to a small extent, and it primarily relies on the reorganization of ECs and the invasion of EPCs [8].

Capillary growth and proliferation are not common phenomena in normal adult tissues, except during wound healing and cyclical changes within the tissues of the female reproductive system (e.g., ovulation and menstruation), but angiogenesis and neoangiogenesis may be initiated with appropriate stimuli. Angiogenesis in adults is typically initiated in response to tissue hypoxia by the release of hypoxia-inducible factors (HIFs), predominantly HIF-1α, which directly increase the expression and/or release of angiogenic stimulators, including vascular endothelial growth factor (VEGF), vascular endothelial growth factor receptor-1 (VEGFR-1), and tyrosine kinase with immunoglobulin-like and EGF-like domains 2 (TIE-2) receptors [9]. Because there is extreme hypoxia in the microenvironment of tissues surrounding tumors, the proliferation of tumor masses may also be indirectly dependent on HIF-1α, which enhances angiogenesis and increases the supply of oxygen to the area of malignant-cell transformation [10,11,12,13]. Therefore, in clinical oncology, the hypoxic microenvironment translates into the occurrence of metastasis and poor prognosis [12,14,15].

During physiological and pathological angiogenesis, EC activation is the first process that occurs [16,17]. The activation of ECs results in the degradation of the vascular basement membrane (VBM), followed by vascular sprouting or vascular splitting, depending on the dominant share of ECs or EPCs, respectively. During this sprouting/splitting phase, transmembrane receptors that help cell–cell and cell–extracellular matrix (ECM) adhesion, the integrins avβ3 and avβ5, play key roles in EC/EPC proliferation, the migration of cells along a gradient of ECM-bound chemoattractants (i.e., haptotaxis), and survival [18,19,20].

Angiogenesis is tightly regulated by the balance between pro-angiogenic and anti-angiogenic factors, including various cytokines and cytokine profiles [21,22]. Tumor angiogenesis is a consequence of disturbances in cytokine balance, and it differs significantly from physiological angiogenesis. The abnormal structure of vessels in cancer is accompanied by altered interactions between ECs and mural cells (pericytes and vascular smooth muscle cells), altered blood flow, increased permeability, and delayed maturation [23].

The inflammatory response is also an important factor affecting angiogenesis [24,25,26]. Inflammation is defined as the homeostatic response of vascularized tissue to a sub-lethal damaging agent that destroys or inactivates invading pathogens, removes waste and debris, and permits the restoration of normal function via resolution or repair [27,28]. Analogous to angiogenesis, the initiation and resolution of the inflammatory response involves the complex and precise coordination of the expression of many factors, including cytokines/chemokines, growth factors, enzymes (mainly proteases), oxidative stress products, and lipid mediators [29,30]. Angiogenesis initiation is often associated with increased capillary permeability, and vascular permeability is greatly increased in acute and chronic inflammation, cancer, and wound repair [31]. This hyperpermeability is mediated by acute or chronic exposure to vascular permeabilizing agents, particularly vascular permeability factor/vascular endothelial growth factor (VPF/VEGF, VEGF-A) [32]. Inflammatory and angiogenic processes overlap, which is why inflammation promotes angiogenesis, and new vessels may support or enhance tissue inflammation [30]. Vascular endothelial cells are a type of innate immune cell that are dependent on pathological conditions. This interaction explains why inflammation contributes to tissue proliferation, tumorigenesis, metastatic spread, and disordered tissue perfusion [31].

### 1.2. Chemokines

Chemokines are small cytokines with chemotactic properties encoded by a large gene family with at least 45 members in humans, and they play important roles in inflammation and angiogenesis [33]. Chemokines are classified based on their primary amino acid sequence and the arrangement of specific structurally important conserved L-cysteine (C) residues at the N-terminus within the mature protein. Variation in the precise configuration of the two cysteines closest to the N-terminus determines the division of cytokines into four classes (subfamilies): C, CC, CXC, and CX3C [34].

Chemokines activate their target cells by signaling via seven (pass)-transmembrane G protein-coupled receptors (GPCRs), which are further divided into conventional chemokine receptors (cCKRs) and atypical chemokine receptors (ACKRs) [35,36]. The promiscuity of chemokines and chemokine receptors on the cell surface, combined with biased signaling (also known as agonist-directed trafficking or functional selectivity) and allosteric modulation of receptor activation, ensures tightly controlled recruitment and positioning (directional migration) of individual cells within the local environment at a given time [37].

Homeostatic chemokines are constitutively expressed under physiological conditions and play a role in cell migration and homing. The local secretion of inflammatory chemokines is rapid and dynamic and aims to recruit effector cells to inflamed tissues [38]. Therefore, chemokines play a key role in the development and homeostasis of the immune system and all protective and destructive immunological/inflammatory responses by participating in the promotion of the activation, migration, differentiation, proliferation, and apoptosis of immune cells. Chemokine signaling and the associated chemotaxis of various cell populations play important roles in the tumor microenvironment (TME) [39,40,41]. This role translates into phenomena directly related to tumor immunology and angiogenesis within the TME, which undoubtedly influence tumor progression and metastasis.

The chemokine CX3CL1 (fractalkine, FKN) deserves special attention in the context of inflammatory angiogenesis and tumorigenesis. CX3CL1 is the only member of the CX3C class of chemokines, and it has well-documented roles in ECs. The uniqueness of the CX3CL1 molecule results from combining the properties of a chemoattractant with adhesive activity [42,43,44,45].

This review presents the role of the FKN signaling pathway in the context of angiogenesis in inflammation and cancer. Information on the participation of cytokines other than FKN in angiogenesis, including cancer angiogenesis, can be found in other, although few, review papers [22,33,46,47,48,49].

## 2. Chemokine CX3CL1 (Fractalkine, FKN)

Two independent teams of researchers from the United States discovered the presence of chemokine (C-X3-C motif) ligand 1 (CX3CL1), but named this molecule fractalkine and neurotactin, in 1997 [50,51]. Baran et al. detected CX3CL1 by searching for chemokine-like sequences in an expressed sequence tag database in the National Centre for Biotechnology Information (NCBI). Bazan et al. chose the name “fractalkine”, which refers to the primary structure of the CX3CL1 molecule, where repeating subunits (fractals) recapitulate the whole. Pan at al. named the CX3CL1 molecule neurotactin after its isolation via the sequencing of the mouse choroid plexus, but it did not remain in common use because it was misleading and referred to a surface glycoprotein previously discovered in Drosophilia melanogaster [52]. Imai et al. (1997) identified a functional high-affinity CX3CL1 receptor, named V28, which was later renamed CX3CR1 [42]. Signaling through CX3CR1, which is the only known FKN receptor with documented function, is responsible for the adhesive and chemoattractant properties of CX3CL1 [42,43,53,54,55].

### 2.1. CX3CL1 Structure

Due to the unique structure of the FKN molecule, the CX3C subclass of chemokines is characterized by a three-amino-acid spacing between the first two conserved L-cysteine residues within its chemokine domain [43,56]. Synthesized as a type I transmembrane protein, FKN exists in two forms, a full-length membrane-bound form and a soluble cleaved form (sFKN), which are generated under physiological conditions via disintegrin and metalloproteinase domain-containing protein 10 (ADAM10) [57,58,59,60]. Stress factors, such as tumor necrosis factor alpha (TNF-α) converting enzyme (TACE or ADAM17), matrix metalloprotease-2 (MMP-2), or lysosomal cathepsin S (CTS), which act locally under conditions of disturbed homeostasis, may also contribute to the formation of sFKN [61,62,63].

The precursor compound of membrane-anchored FKN is a polypeptide with 397 amino acid residues and contains a 24-amino-acid signal peptide (SP) [61]. Mature (SP-free) transmembrane FKN consists of 373 amino acids, which form an extracellular N-terminal (chemokine) domain (aa 76), a mucin-like stalk (aa 241), a transmembrane α helix (aa 19), and a short intracellular domain (aa 37) in the form of a cytoplasmic tail [57,64,65]. The total molecular weight of FKN is approximately 17.5 kDa, but it is 95 kDa after glycosylation [50,51]. Typically composed of 317 amino acids, sFKN consists of a chemokine domain and an extracellular mucin-like stalk, which weigh approximately 14.7 kDa and 80 kDa after glycosylation, respectively [57,66,67]. However, there are some inconsistencies in the molecular weight and amino acid content of sFKN, which is likely due to the multiple forms of soluble CX3CL1 generated through shedding from the cell surface at alternative sites [68,69]. The molecular structures of both forms of FKN are shown in Figure 1.

### 2.2. CX3CL1 Function

The membranous and soluble forms of CX3CL1 (mFKN and sFKN, respectively) exhibit different functions, although both signal through the CX3CR1 receptor, which is a class of GPCR within the superfamily of seven transmembrane-spanning proteins that respond to a diverse range of chemical and sensory stimuli [70].

#### 2.2.1. Membrane-Bound CX3CL1 (mFKN) as an Adhesion Molecule

Notably, the chemokines CXCL16 and FKN are the only chemokines that bind directly to the cell membrane via the transmembrane domain and mucin-like stalk [71,72,73]. FKN synthesis occurs primarily in ECs, which means that this chemokine has direct access to leukocytes in the bloodstream. Therefore, membrane-bound FKN mediates strong adhesion via binding to CX3CR1 on leukocytes [74,75]. This strong interaction between FKN and CX3CR1-expressing blood cells may result from a very low dissociation rate of this endothelial–white blood cell adhesion that may support the trans-endothelial migration of leukocytes during inflammation [76,77,78].

FKN is an adhesion molecule that mediates leukocyte adhesion directly or in cooperation with other tethering proteins, such as cadherins, immunoglobulin superfamily cell adhesion molecules (e.g., cluster of differentiation 106 [CD106] or vascular cell adhesion protein 1, also known as vascular cell adhesion molecule 1 [VCAM-1]), selectins, and syndecans [43,79,80,81]. The indirect effect of FKN via cell adhesion molecules was also investigated. By counteracting the shear stress forces in most vascular beds, FKN does not recruit leukocytes alone because it does not provide optimal adhesion strength [61,81]. The significant adhesion of leukocytes to FKN peaks at 2 dynes/cm^2^ but is minimal at 10 dynes/cm^2^. Contrary, VCAM-1 recruits cells from whole blood at 10 dynes/cm^2^. However, when acting together, FKN and VCAM-1 show synergistic effects and cause a twofold increase in the number of adherent cells compared to VCAM-1 alone, which suggests that FKN mediates adhesion at high shear when combined with a molecule that mediates leukocyte tethering [78].

FKN is a multi-domain transmembrane chemokine that causes leukocytes to adhere without rolling and migration by sharing its chemokine domain (CD) to CX3CR1 [82]. Other domains of the FKN molecule also have key functional importance, which shows that the molecular structure of FKN may be precisely adapted to capture CX3CR1 in circulating cells. For example, the mucin stalk (mucin-like domain) holds and presents the CD away from the cell membrane surface, and its stiffness is achieved via a high degree of glycosylation. Therefore, the adhesion potential of FKN may be limited to a greater extent by the shortening of the mucin stalk of CX3CL1 and mutation of the potentially glycosylated residues of the mucin stalk than the absence of the cytosolic domain (cytoplasmic tail) [83]. The cytosolic domain is responsible for the robustness of adhesion via the cytoskeleton. FKN is present as a monodisperse bundle on the surface of CX3CR1-expressing cells, and its packing is driven by its transmembrane domain (transmembrane α-helical region), which initiates the stabile aggregation of an adequate amount of monomers to guarantee adhesion and prevent rolling [73].

The membrane compartment of FKN mediates a special dynamic equilibrium between the plasma membrane and intracellular vesicular trafficking from the intracellular compartment. Therefore, the constitutive internalization of pre-synthesized FKN molecules, which prevents FKN degradation by cell surface metalloproteases and accelerates the mobility of its intracellular content, occurs [84,85]. The two-adapter protein 2 (AP2)-binding motifs that bind clathrin, which plays a major role in the formation of coated vesicles, are crucial for the endocytosis-based internalization of FKN: YQSL is located within amino acid residues 362–365 (the cytoplasmic tail), and YVLV is located at positions 392–395 (the precursor form of FKN) [84]. The spatial distribution of FKN in individual subcellular compartments is also clearly influenced by the properties of vesicle-associated v-soluble N-ethylmaleimide-sensitive factor attachment protein receptor (SNARE) proteins, such as syntaxin 13 (STX13) and vesicle-associated membrane protein 3 (VAMP3) [86].

#### 2.2.2. Soluble CX3CL1 (sFKN) as a Chemoattractant

Soluble FKN is cleaved via proteolysis by metalloproteinases, and it contains a chemokine domain (CD). SFKN exhibits functions typical of conventional chemokines and chemotaxis. As in many chemotactic cells, the signal regulating motility is initiated via the binding of sFKN on the cell surface to a GPCR class receptor, to which CX3CR1 belongs [43,87,88]. The main tasks of chemokines in the body are the modulation and targeting of the immune response implemented via chemotactic effects on leukocytes by creating a concentration gradient [89,90,91,92]. The versatility and diversity of the chemotactic effects of sFKN occur because CX3CR1 is expressed constitutively or involved in inflammatory responses in various cells, including hemato- and non-hematopoietic lines. The former also includes blood cells circulating in the vascular system, such as CD4+ and CD8+ T cells, B cells (CD19+), natural killer (NK; CD56+CD3-) cells, monocytes (CD14+), thrombocytes, dendritic cells (CD11c+), mast cell (MC) progenitors, peripheral blood-derived hematopoietic stem cells (PBHSCs), and neutrophils, to a much lesser extent [93,94,95,96,97]. The weak expression of CX3CR1 in neutrophils may explain why sFKN is not a major chemoattractant for the migration of neutrophils across the microvascular endothelium [98,99]. Notably, the very weak or ineffective effect of sFKN on stimulating neutrophil migration is not determined by CX3CR1 expression per se. Imai et al. [42] showed that although 80% of CD14+ monocytes expressed CX3CR1, only 1% of the input cells, i.e., only 1.3% of the receptor-expressing cells, migrated to sFKN. T helper-1 cells, a monocytic cell line, migrated to monocyte chemoattractant protein-1 (MCP-1) but not to sFKN, although these cells expressed the FKN receptor on their surface. Therefore, membrane-bound FKN, which is present predominantly in vascular ECs, efficiently mediates the binding and adhesion of neutrophil populations [42,58].

Trans-endothelial migration assays estimate the movement of sFKN-stimulated leukocytes through the endothelial cell layer, and the contribution of sFKN to angiogenesis, including tumorigenesis, tumor metastasis, and EC chemotaxis, has been demonstrated [100,101,102].

## 3. CX3CR1—The Sole Fractalkine Receptor

The direct biological actions of both forms of fractalkine, adhesive for membrane-bound FKN and chemotactic for sFKN, are the result of interaction with the dedicated CX3C motif chemokine receptor 1 (CX3CR1, previously designated V28), also known as the fractalkine receptor or G protein-coupled receptor 13 (GPR13) [42,43,53,54,55]. Noticeably, the existence of the sole receptor for CX3CL1 makes it much easier to interpret the observed biological effects related to this chemotactic cytokine with respect to the CX3CL1/CX3CR1 axis.

### 3.1. CX3CR1 Structure

The genome of the CX3CR1 gene in humans is located on the short arm of chromosome 3 (3p22.2). CX3CR1 is composed of six exons (only two contain coding regions), and three intronic elements and three promoters are involved in the regulation of its genomic sequence [73,103]. CX3CR1 is evolutionarily conserved and encodes identical or similar sequences (four transcript variants) in mice and rats, despite the different locations of CX3CR1 (on chromosome 9 [9qF4] in mice and on chromosome 8 [8q32] in rats) [103,104,105].

The sequence of the 355 amino acids and the spatial structure of the transmembrane protein (MW∼40 kDa) constituting CX3CR1 are well known [42]. The binding of CX3CR1 to metabotropic receptors within the most numerous class A (rhodopsin-like receptors) in the GPCR family of proteins, which are composed of a monomeric protein containing an extracellular domain with a signaling ligand binding site and an intracellular domain binding the G protein, occurs [106]. A structural diagram of CX3CR1 is shown in Figure 2.

The polypeptide chain of CXC3CR1 is composed of seven α-helical structures extending across the cell membrane (transmembrane segments or domains: TM1–TM7) and exceeding its thickness in both directions, i.e., into the extracellular space and the cytoplasm. This conformation creates eight amino acid chains on both sides of the cell membrane that connect individual TMs: three extracellular loops (ECL1–ECL3), another three intracellular loops (ICL1–ICL3), and two linear chains forming an extracellular amino terminus (NH2) and an intracellular carboxyl terminus (COOH) at the ends of the molecule [65,114]. The CX3CR1 molecule contains a disulfide bond connecting two conserved cysteine (C) residues located at the top of the extracellular side of TM3 and within ECL2 [109].

There are binding sites within the ECLs and NH2 terminus for functional ligands, such as CX3CL1 (FKN) and CCL26 (eotaxin-3); antibodies; and some pathogens, such as bacteria and viruses [115,116,117,118]. The appropriate level of tyrosine (T) sulfation at the NH2-terminus is required for maintaining the normal activity of most GPCR receptors for chemokines [61,119].

ICL2 is of key functional importance on the other side of the plasma membrane because it contains the canonical DRYLAIV motif, which is composed of a sequence of seven amino acids with three-letter abbreviations: Asp-Arg-Tyr-Leu-Ala-Ile-Val [104,110]. This protein sequence motif provides a docking point that is essential for the coupling of CX3CR1 to the heterotrimeric G protein, which belongs to the Gαi family. This binding is crucial for the induction of classical signaling pathways because metabotropic receptors do not contain ion channels in their structure and only influence ion flow by activating the intermediary G protein [61,120]. The binding of an agonist to CX3CR1 causes conformational changes in the receptor with the subsequent dissociation of the components of the heterotrimeric G complex, which consists of alpha (α), beta (β), and gamma (γ) subunits. The binding of guanosine diphosphate (GDP) allows the α subunit to bind to the β and γ subunits to form an inactive trimer (inactive Gα-GDP state). The binding of an extracellular signal (ligand) to CX3CR1 enables the G protein to bind to the receptor and causes GDP to be substituted by guanosine triphosphate (GTP) to create the active Gα-GTP state [114,120,121]. The COOH-terminal serine residues (S) are susceptible to G protein-coupled receptor kinase (GRK)-mediated phosphorylation and subsequent desensitization. However, more recently characterized functions of GRKs as scaffolds and signaling adapters suggest that this small family of proteins modulates CX3CR1 activity in a more complex way [122].

Similar to other chemokine receptors, CX3CR1 exhibits polymorphisms, which may be responsible for its varying affinity for CX3CL1 and other ligands [123]. The polymorphic residues at positions 249 and 280 may be responsible for dysfunctional CX3CR1 variants, including variants identified in various cancers [112,113]. The polymorphism of CX3CR1 is also related to diseases of the cardiovascular system (e.g., atherosclerosis) and nervous system (e.g., Alzheimer’s disease) and infections (e.g., systemic candidiasis) [124,125,126,127].

### 3.2. FKN Signaling via CX3CR1

#### 3.2.1. Conformational Rearrangements following FKN Binding

Metabotropic receptors, including CX3CR1, constitute the largest family of cell surface proteins involved in signaling across biological membranes, and conformational changes after ligand attachment are crucial for initiating intracellular signaling pathways [114,121]. Under physiological conditions, the sole endogenous CX3CR1 ligand, FKN, binds to the orthosteric sites of the receptor [77]. The affiliation of CX3CR1 with the A1 subfamily within the rhodopsin-like GPCRs, established on the basis of phylogenetic analysis, indicates that the receptor molecule resembles the structure of rhodopsin and transduces extracellular signals via interaction with guanine nucleotide binding (G) proteins [128,129]. Given a predictable structure, conservation of a few amino acids in the region crucial for G protein activation, and activation by a small ligand, rhodopsin is a model compound for the assessment of conformational changes associated with the activation of subclass A GPCRs [130,131]. Conformational changes in CX3CR1 after ligand binding were also examined indirectly using US28, which is a virus-encoded GPCR showing 29% sequence identity with CX3CR1 [132,133]. US28 also binds to CX3CL1 and acts during human cytomegalovirus infection [134].

The term “seven-transmembrane (7TM) receptors” is often used interchangeably with GPCRs and reflects their seven membrane-embedded helices and additional signaling independent of G proteins [135,136].

Despite resolving the spatial structure of many complexes of chemokines in classes other than FKN with proper receptors, our knowledge of specific chemokine recognition mechanisms in the CX3C subfamily remains incomplete [133,137]. Creating models of the crystal structure and the use of cryo-electron microscopy (cryo-EM) to study US28-FKN and US28-engineered FKN (chemokine CX3CL1.35) complexes has been insufficient to explain the conformational changes in CX3CR1 that occur with its activation in humans [138,139]. However, cryo-electron microscopy data indicate the involvement of cholesterol in the regulation of CX3CR1 activation, which translates into conformational changes in the CX3CR1-Gαi complex observed in ligand-free and CX3CL1-bound states at 2.8- and 3.4-Ångström (Å) resolutions, respectively [114,121]. A comparison of the overall structures of the CX3CR1-Gα and CX3CR1-CX3CL1-Gαi complexes revealed that despite exhibiting almost the same conformations by these two complexes with a receptor Cα (residues T31 to Y305) root mean square deviation (RMSD) of 1.4 Å, the superposition of CX3CR1 in the two states reveals distinct differences in the extracellular region of the receptor. Compared to the ligand-free complex, the N-terminus of the activated receptor moves much more strongly toward the center axis of the helical bundle, and extracellular loop 2 (ECL2) is repelled upon CX3CL1 binding. [121]. The movement outside the cell, shown by helix VI and treated as a specific “conformational marker” of activation within the previously structurally solved representatives of class A GPCRs, is clearly smaller for the CX3CR1-CX3CL1-Gαi complex [140]. Initially, ligand-free CX3CR1 shows a more central helix VI location. For example, it shows the only 2.3 Å outward movement of the intracellular end of helix VI, while the range of this movement is 8.2 and 6.7 Å in the active structures of C-C chemokine receptor type 5 (CCR5)-Gαi and US28-Gαi, respectively. Therefore, the different conformations of CX3CR1 cause its Gαi-coupling surface area to be 900 Å2, which is much larger than those of the CCR-Gαi (826 Å2) and US28-Gαi (790 Å2) complexes. Therefore, the distinct conformations of CX3CR1 and the Gαi coupling interface suggest the existence of alternative activation mechanisms of CX3CR1 and provide some insight into the diversity of G protein-dependent intracellular signaling of FKN [121,141,142]. This activity is complemented by the fact that activated CX3CR1 exhibits conformational rearrangements of key conserved activation motifs in class A GPCRs, including the canonical DRYLAIV motif, which provides a docking point for the G protein [38,143,144].

Natural variants of CX3CR1 may differ conformationally during activation, and conformational changes in CX3CR1 accompanying activation may be disrupted as a result of mutations and may be associated with various diseases, including changes in the risk of several cancers [114,145,146,147,148,149,150]. Glutaminyl cyclase (QC)-catalyzed N-terminus pyroglutamate (pGlu) formation of the ligand (FKN) may determine the stability or interaction with CX3CR1, and it is, therefore, essential for the full biological activity of FKN [151].

In addition to conformational changes in activated CX3CR1, cells expressing CX3CR1 (e.g., microglia and lymphocytes) undergo actin polymerization and cytoskeletal rearrangements, which are necessary to initiate chemotaxis [152,153,154,155].

#### 3.2.2. Main FKN/CX3CR1 Signaling Pathways

The binding of membrane-bound FKN and the soluble form of cleaved FKN to CX3CR1 at its extracellular determinants localized within the amino terminus and the third extracellular loop (ECL3; see Figure 2) independently contributes to and is a necessary condition for proper conformational rearrangements preceding the activation of heterotrimeric G proteins associated with CX3CR1 [156]. This binding of the agonist to the receptor involves two steps. Step one comprises high-affinity FKN binding involving Tyr14, Asp25, and Glu254. This initial interaction then leads to the inclusion of Glu13, Asp16, and Asp266 (step two) [156,157]. On the other side of the cell membrane (inside the cell), the Gα subunit disassociates from the membrane and interacts with G protein regulatory (GPR) domain-containing proteins. RIC8 (synembryn), which is a non-receptor guanine nucleotide exchange factor for Gα subunits, facilitating the exchange of GTP for GDP [158,159]. The presence of active GTP-bound Gα in the G protein complex leads to its dissociation into Gαi-GTP and a GβGγ dimer. Activated Gαi interacts with downstream effectors [160].

The resulting FKN-CX3CR1 axis transduces several well-characterized signaling pathways that lead to the activation of several transcription factors (e.g., signal transducer and activator of transcription protein [STAT], nuclear factor kappa-light-chain-enhancer of activated B cells [NF-κβ], and cAMP/Ca2+ response element binding protein [CREB]) and the inhibition of other factors (e.g., members of the class O forkhead box transcription factor [FOXO]) [44,106,161]. Most of these CX3CR1 signaling pathways are shared with other chemokine receptors, including the following:

❶ The stimulation of calcium mobilization from intracellular stores via the phospholipase C (PLC)/protein kinase C (PKC) pathway [162,163], leading to the activation of the respective kinases, resulting in subsequent downstream signaling within ❷ the Janus kinase (JAK)/STAT pathway, ❸ the phosphoinositide 3-kinase (PI3K)/protein kinase B (Akt)/IkappaBeta (Iκβ) kinase (IKK)/Iκβ/NF-κβ pathway, ❹ Ras kinases (Ras)/Raf kinases (Raf)/mitogen-activated protein kinase kinase (MEK)/extracellular signal-regulated kinase (ERK), and ❺ the MEK kinase (MEKK)/cJun NH(2)-terminal kinase (JNK)/CREB or MEKK/mitogen-activated protein kinases (P38)/CREB pathways [100,164,165,166].

The most important signaling pathways within the FKN/CX3CR1 axis are presented in Figure 3.

Both forms of FKN, the membrane-bound and—resulting from membrane shedding by lysosomal protease cathepsin S (CTS) and/or metalloproteases (a disintegrin and metalloproteinase domain-containing protein 10 [ADAM10], a disintegrin and metalloprotease 17 [ADAM17], also called tumor necrosis factor alpha converting enzyme [TACE], and matrix metalloprotease-2 [MMP-2])—soluble FKN, activate the same signaling pathways promoting adhesion or chemotaxis, respectively [156,167]. The presence of an active, guanosine triphosphate (GTP)-bound Gα in the G protein complex leads to dissociation into Gαi-GTP and a GβGγ dimer. Once activated, Gαi can go on to interact with downstream effectors [160]. The effect on gene transcription is achieved by activating the signal transducer and activator of the transcription protein (STAT), nuclear factor kappa-light-chain-enhancer of activated B cells (NF-κβ), and cAMP/Ca2+ response element binding protein (CREB) while inhibiting the members of the class O of forkhead box transcription factors (FOXO) [44,106,161]. The involvement of the phospholipase C (PLC)/protein kinase C (PKC) pathway in the intracellular divalent calcium cation (Ca2+) mobilization that may influence chemotaxis is well documented [162,163]. By inhibiting the expression of pro-apoptotic proteins and FOXO activity, signaling pathways via phosphoinositide 3-kinase (PI3K)/protein kinase B (Akt)/IkappaBeta (Iκβ) kinase (IKK)/Iκβ/NF-κβ pathways, Ras kinases (Ras)/Raf kinases (Raf)/mitogen-activated protein kinase kinase(MEK)/extracellular signal-regulated kinase (ERK) and MEK kinase (MEKK)/cJun NH(2)-terminal kinase (JNK)/CREB, or MEKK/mitogen-activated protein kinases (P38)/CREB pathways can increase the survival of cells, including cancer cells [168,169,170].

Other abbreviations include the following: AC—adenylyl cyclase; cAMP—cyclic adenosine monophosphate; Gα, Gβ, and Gγ—subunits of the heterotrimeric G proteins (G protein complex); Gαi—activated Gα subunit of the G protein complex; GDP—guanosine diphosphate; IP3—inositol 1,4,5-trisphosphate; JAK—Janus kinase; PIP2—phosphatidylinositol 4,5-biphosphate; RIC8—a non-receptor guanine nucleotide exchange factor for Gα subunits (also known as synembryn)

One of the most commonly understood signals generated by the FKN/CX3CR1 pathway is the prevention of apoptosis, primarily in monocytes. FKN/CX3CR1 signaling induces the expression of anti-apoptotic genes, primarily BCL2 and BCL-xL (B-cell lymphoma extra-large) [171]. Reduced expression of pro-apoptotic proteins and FOXO promotes cell survival and cancer cells in certain conditions [168,169,170].

Metabotropic CX3CR1 is not isolated within the biological membrane. Therefore, many signaling pathways activated by FKN depend on the functional state of other receptors, including epidermal growth factor receptor (EGFR), a member of the family closely related to receptor tyrosine kinases ErbB-1 (EGFR) and ErbB-2 (HER2/neu) [172,173]. EGFR is involved in cell signaling pathways that control cell division and survival [174,175,176], whereas CX3CR1-dependent intracellular signaling cascades are responsible for the processes of migration and proliferation based on increased cell survival [100,160,161,162]. Mutations of the EFGR gene located at the short arm of chromosome 7 (7p11. 2) affect its expression or activity and may contribute to the development of many cancers, with CX3CR1 acting as one of the transactivators of ErbB-1 and ErbB-2 (see Chapter 5 on the FKN/CX3CR1 axis and tumorigenesis) [102,177,178,179,180,181]. Transactivation, which is well characterized for EGFR, represents the process whereby GPCRs activate receptor tyrosine kinases (RTKs), most of which are receptors for numerous neurotrophic factors and growth factors, with the subsequent activation of the signal paths (downstream signaling), such as mitogen-activated protein kinase (MAPK) [182,183].

A significant advance in CX3CR1 research was the demonstration that FKN exhibits an autoregulatory function and may induce its own expression via the activation of pertussis toxin-sensitive G proteins, PI3K, phosphoinositide-dependent kinase 1 (PDK1), Akt, NIK, IKK, and NF-κβ. Tumor necrosis factor alpha (TNFα) plays a key role in this autoregulatory loop because it induces the expression of FKN and CX3CR1 in an NF-κβ-dependent manner [43,184]. CX3CR1 autoregulation occurs via the use of NF-κβ inhibitors, and a reduced FKN concentration is accompanied by increased CX3CR1 expression [185].

## 4. FKN/CX3CR1 Axis and Inflammation

As mentioned in Section 2.2.2., CX3CR1 is a chemoattractant of soluble CX3CL1 (sFKN) that is constitutively expressed or induced via inflammation in many hemato- and non-hematopoietic lines [93,94,95,96,97]. The differential degree of CX3CR1 expression throughout the body applies to immune and non-immune system cells, primarily in a cell-type-specific manner [74,186].

The activation of the FKN/CX3CR1 axis suggests the occurrence of immune cell chemotaxis, which is determined by ligand concentration gradients [53,113]. Because cells expressing CX3CR1 are involved in inflammatory and anti-inflammatory responses, the final effect of this chemotaxis is dependent primarily on local environmental conditions. This effect constitutes a specific duality of action of FKN, which may facilitate the maintenance of homeostasis, but FKN may play a key role in the pathomechanism of inflammation in pathological conditions [77,127,187,188]. However, there is growing evidence that the impact of CX3CR1 may be tissue- and disease-specific [74]. Notably, in addition to inducing the adhesion and chemotaxis of leukocytes, FKN has anti-apoptotic effects and increases the average survival of multiple cell types, both during homeostasis and inflammation [189,190,191].

The involvement of FKN and FKN/CX3CR1 signaling under physiological conditions and the pathological inflammatory responses in selected tissues/organs/systems are summarized in Table 1.

### 4.1. FKN/CX3CR1 Axis and Inflammation-Induced Angiogenesis 

#### 4.1.1. Interdependence of Inflammation and Angiogenesis

Angiogenesis, i.e., the formation of new blood vessels from pre-existing vessels via sprouting or splitting is fundamentally important in body development and tissue regeneration [271,272]. Phenomena accompanying angiogenesis, such as the migration, growth and differentiation of endothelial cells, are tightly controlled primarily by cytokines that use multiple signaling pathways. Notably, angiogenesis after the completion of individual development primarily occurs in conjunction with a chronic inflammatory response [273]. Chronic inflammation and hypoxia, which are the principal physiological stimuli that induce angiogenesis, clearly coincide [274,275]. Therefore, inflammation and angiogenesis, including tumor angiogenesis, are two interdependent processes that may enhance each other [272,275]. Most angiogenic factors have a functional duality, consisting of pro-inflammatory and pro-angiogenic effects [224,276,277,278]. The process of initiating vessel formation itself is most often associated with an initial increase in the permeability of the microcirculatory vessel wall, which allows angiogenic factors contained in the plasma to penetrate into the interstitial compartment. After the destabilization of the endothelial cell monolayer, the directional motility (haptotaxis) of these cells occurs toward angiogenic stimuli within the extravascular space [279]. Integrins (avβ3 and avβ5) play important roles in this process and determine adhesion to matrix proteins, with the concomitant proliferation of ECs lining the vessel wall occurring in order to replace previously migrated cells. Neuropilins (NRP-1 and NRP-2), which are highly conserved type I membrane glycoproteins that act as co-receptors for vascular endothelial growth factors (VEGFs) together with VEGF receptors (VEGFRs), also play important roles in this stage of angiogenesis. High levels of NRP1 can be identified on the arterial endothelium, but NRP2 expression is limited to lymphatic and venous endothelial cells [280,281]. For example, NRP-1 binding to several VEGF-A isoforms promotes the interaction of growth factors with VEGFR-2, which increases receptor phosphorylation. NRP-1 is required for EC adhesion to soluble VEGFR-1 [282]. NRP-2 is an important angiogenic player in the promotion of EC migration and adhesion by regulating integrin alpha 5 (ITGA5/CD49e) recycling [283]. NRP-2 may also act as an inflammation-sensing protein and is rapidly and dramatically induced in myeloid cells, especially macrophages, under inflammatory conditions [284]. In response to hypoxia, there is a local increase in hypoxia-inducible factor-1 alpha (HIF-1α) production, which likely controls angiogenesis and metabolism by upregulating hypoxia-induced genes, such as the interleukin-33 (IL-33) gene and VEGF gene [285].

During migration and proliferation, ECs form cord-like structures in target tissues that later canalize to form fully functional vessels, which are further supported by surrounding pericytes [286]. Tight cell–cell adhesion is a consequence of the expression and function of different adhesion molecules, such as platelet–endothelial adhesion molecule-1 (PECAM-1, CD31) and vascular–endothelial cadherin (VE-cadherin, CD144) [287,288]. Newly formed vessel network stabilization requires remodeling, hierarchization, and quiescence [279,289].

The proper course of angiogenesis requires precise coordination at the level of many signaling pathways, which allows for a change in the dynamic balance between angiogenesis inhibitors and stimulators (anti- and pro-angiogenic factors) toward the latter [16,290]. Clinical observations and histopathological data confirm that the angiogenesis coexisting with chronic inflammation tends to prolong and intensify the inflammatory response [288,291]. Numerous pathological conditions, such as inflammatory bowel disease (IBD), cancer growth and tumor metastasis, arthritis, diabetic retinopathy, and ischemic cardiovascular diseases (including stroke), are associated with abnormal inflammatory angiogenesis [292,293,294,295,296].

#### 4.1.2. Pro-Angiogenic Effects of FKN/CX3CR1 Signaling on the Inflammatory Response

Coincidental inflammation and hypoxia are generally found in chronic inflammation. Both of these conditions contribute to the activation and/or potentiation of the NFĸB gene regulator. This effect is important for the activity of the FKN/CX3CR1 axis because NF-κB is a main controller of FKN expression [297]. An increase in NF-ĸB activity in chronic inflammation is induced by the action of many pro-inflammatory cytokines, including TNFα and IL-1 [298,299]. The stimulation of cluster of differentiation 40 (CD40, also known as TNFRSF5—tumor necrosis factor receptor superfamily member 5), which is a costimulatory protein expressed on antigen-presenting cells (APCs), by CD40 ligand (CD40L) expressed on CD4 T lymphocytes (helper T cells) shortly after activation induces the classical and alternative NF-κB pathways in endothelial cells and promotes CX3CL1 expression [300,301]. The intracellular signaling pathways leading to increased FKN expression via increased NF-ĸB synthesis include the activation of phosphoinositide-3-kinase (PI3K). An autoregulatory mechanism in which FKN controls its own expression in an inflammatory environment also involves the PI3K/Akt/IKK/IKβ/NF-ĸB signaling pathway [184]. The Akt signaling-related activations of the vasodilation pathway via eNOS activation and increased NO production in the endothelium are also closely associated with angiogenesis [302,303].

In addition to the PI3K/Akt/IKK/IKβ/NFĸB pathway, the pro-angiogenic effects of FKN/CX3CR1 axis activation by vascular ECs are mediated via the PKC/Ras/Raf/MEK-ERK or PKC/MEKK/MEK/ERK signaling pathways [168]. A two-step sequence of events then occurs: HIF-1α is upregulated and phosphorylated in hypoxia via an ERK-dependent pathway, with the subsequent enhancement of VEGF-A gene transcription in monocytes [304,305,306]. VEGF-mediated angiogenesis requires NO production from activated endothelial NO synthase (eNOS) because sprouting angiogenesis requires vasodilation and increased vascular permeability [307,308].

The pro-angiogenic effects of FKN/CX3CR1 signaling on the inflammatory response are shown in Figure 4.

Membrane-bound FKN (mFKN), which determines adhesion, and soluble FKN (sFKN) associated with chemotaxis are involved in the activation of the CX3CR1 signaling pathway [156,167]. Once activated, CXCR1 interacts with downstream effectors, ultimately leading to angiogenesis accompanied by vasodilation and resistance to apoptosis [160]. The pro-angiogenic effects of the activation of the FKN/CX3CR1 axis are mediated by phosphoinositide 3-kinase (PI3K)/protein kinase B (Akt)/IkappaBeta (Iκβ) kinase (IKK)/Iκβ/nuclear factor kappa-light-chain-enhancer of activated B cells (NF-κβ), as well as by the protein kinase C (PKC)/Ras kinases (Ras)/Raf kinases (Raf)/mitogen-activated protein kinase (MEK)/extracellular signal-regulated kinase (ERK) or PKC/MEK kinase (MEKK)/MEK/ERK signal pathways [168]. The activation of PI3K intracellular downstream signaling is linked to increased NF-ĸB synthesis and secondary, increased FKN expression, whereas PKC … ERK signaling is responsible for the upregulation of hypoxia-inducible factor 1-alpha (HIF-1α). Next, HIF-1α affects its target genes, which results in, among others, increasing VEGF-A transcription in monocytes [304,305]. Vasodilation is related to Akt signaling with subsequent endothelial nitric oxide synthase (eNOS) activation and increased nitric oxide (NO) production [302,303]. Increased cell survival due to resistance to apoptosis is a consequence of Akt and ERK signaling with the inhibition of pro-apoptotic proteins, and in the case of the Akt pathway, also the members of the class O of forkhead box transcription factor (FOXO) inhibition and increased expression of the (anti-apoptotic) B-cell lymphoma (Bcl-2) protein. The involvement of the phospholipase C (PLC)/protein kinase C (PKC) pathway in the intracellular divalent calcium cation (Ca2+) mobilization that may influence chemotaxis is well documented [162,163]. 

## *5.* FKN/CX3CR1 Axis and Tumorigenesis

### 5.1. Hypoxia and Angiogenesis in the Tumor Microenvironment (TME) 

Angiogenesis and tumorigenesis are interconnected. However, the main difference between cancer cells and normal cells is that cancer cells grow uncontrollably beyond the control of the immune system. Because chemotaxis is one of the key phenomena during cancer development, invasion/progression, and metastasis, the presence of chemokines in the tumor microenvironment, especially chemokines with angiogenic properties, is essential [309]. Another characteristic feature of cancer is disturbed cell adhesion processes with significant changes in the functions and quantitative profiles of cell adhesion molecules, which lead to the loss of cell-to-cell adhesion [310].

The essential role of angiogenesis in tumor growth and the spread and establishment of metastases has been well documented [311,312]. Cancer tissue cells that show spheroidal growth in vitro have an upper volume limit determined by the nutrients and oxygen diffusion distance (from the culture medium to the spheroid core) [313,314]. The expansion of tumors growing in tissue in vivo is limited by the nutrient diffusion distance from the nearest capillary, which is 100–500 microns [315,316,317]. Therefore, a further increase in the size of the tumor requires increased vascularization within the tumor tissue, and the process through which tumor-associated neovessels sprout from existing blood vessels is called “tumor angiogenesis” after Folkman [315].

Compared to normal vessels, new vessels produced by most types of cancer cells during tumor angiogenesis have abnormal morphology and function (e.g., increased permeability), leading to limited blood supply efficiency [318]. Paradoxically, abnormal blood and lymphatic vessels create a hostile tumor microenvironment (TME) that is characterized by hypoxia, lowered pH, and elevated interstitial fluid pressure, which noticeably maintain the malignancy of the tumor [319]. Hypoperfusion-related hypoxia makes cancer cells more aggressive, and “leaking vessels” allow these cells to travel to distant sites to metastasize [320]. Hypoxia creates an abnormal TME in which angiogenesis is constantly upregulated and immune system functions are limited (e.g., recruitment of tumor-infiltrating lymphocytes—TILs), leading to tumor-mediated immunosuppression [321]. Hypoxia contributes to the progression of cancer and resistance to photodynamic therapy, chemotherapy, and radiotherapy [322,323,324].

### 5.2. FKN in the TME

As a unique chemokine expressed in many cell types that exhibits chemoattractant (sFKN) and pro-angiogenic effects and the typical features of an adhesion molecule (mFKN), FKN has been intensively studied in the context of carcinogenesis and metastasis [224,252,306]. The presence of a signaling pathway associated with only the metabotropic receptor CX3CR1 facilitates the interpretation of the observed relationships and narrows potential therapeutic targets [44,198,211].

Because each cancer shows considerable heterogeneity (including DNA methylation heterogeneity) depending on the type and cell-specific variability within the same type, receptor expression, chemotaxis, and cellular adhesion are not fully predictable [325,326,327]. This relationship also applies to the function of the FKN/CX3CR1 axis within the TME. It is crucial to precisely define whether FKN/CX3CR1 signaling may be a therapeutic target in a given oncological case and determine whether its inhibition or stimulation is beneficial [88].

#### 5.2.1. FKN/CX3CR1 Signaling May Promote Tumorigenesis

Most publications demonstrated the pro-tumorigenic and pro-metastatic roles of FKN/CX3CR1 signaling across multiple blood and solid malignancies [328]. The promotion of tumor growth and metastatic spread is generally associated with the production of new vessels in response to hypoxia within the TME and, less often, with the chemotaxis of tumor cells. The recruitment of circulating CX3CR1+ monocytes is critical for tumor angiogenesis [329].

For example, CX3CR1 signaling enhances the accumulation of tumor-associated microglia/macrophages and angiogenesis during the progression and malignant transformation of low-grade gliomas. Notably, one study revealed increased survival in patients (n = 45) with the presence of a common CX3CR1 V249I polymorphism, which correlated with reduced tumor vessel density and reduced M2 macrophage infiltration [149].

Studies in hepatocellular carcinoma (HCC), the course of which is inherently associated with the inflammatory process and the upregulation of cytokines, have shown that FKN knockout inhibits the in vitro and in vivo angiogenesis of HCC HepG2 cells [330].

CX3CR1 was expressed in a histological grade- and stage-dependent manner in histopathological samples obtained from patients with colorectal cancer. CX3CR1 upregulation correlated with poor prognosis due to the increased survival of angiogenic macrophages in the TME, which contributed to tumor metastasis [331]. High CX3CR1 expression or high expression of the CX3CL1/CX3CR1 axis were also independent negative prognostic factors in pancreatic ductal adenocarcinoma. CX3CL1 and CX3CR1 expression (77.1 and 66.7%, respectively) was clearly greater in malignant areas than in peritumoral areas [332].

mFKN promotes cell–cell adhesion for communication between tumor cells and vascular ECs and, consequently, angiogenesis. This finding was confirmed by the result showing that the small interfering RNA-mediated knockdown of the FKN gene inhibited melanoma B16-F0 cell growth in vivo, which correlated with decreased angiogenesis around the tumor [333].

The FKN/CX3CR1 interaction may be crucial in the development of prostate cancer metastases to bone tissue. FKN promotes the adhesion of human prostate cancer cells to bone marrow endothelial cells and their migration toward human osteoblasts in vitro [334]. This effect occurs because osteoblasts and stromal and mesenchymal cells derived from human bone marrow express mFKN, whereas sFKN is present in bone marrow supernatants [335]. After malignant transformation, endothelial cells overexpress CX3CR1, but CX3CR1 occurs in minimal quantities in the endothelium of the normal prostate gland. Androgens may promote the extravasation of CX3CR1-bearing cancer cells on an FKN concentration gradient, but their ability to adhere to the bone marrow endothelium is not altered [336]. In vivo animal models showed that the overexpression of CX3CR1 induced the spinal metastasis of prostate cancer via the FKN/steroid receptor coactivator (Src)/focal adhesion kinase (FAK) signaling pathway [337]. Exposure to FKN may result in epithelial-to-mesenchymal transition (EMT) with enhanced CX3CR1+ cell migration, which is symptomatic of increased invasive and metastatic potential during prostate cancer progression. This mechanism of FKN-dependent EMT involves the activation of tumor necrosis factor-α converting enzyme (TACE)/transforming growth factor-α (TGF-α)/epidermal growth factor receptor (EGFR)] signaling [102].

CX3CR1 was more highly expressed in spinal metastases than para-tumor tissue in breast cancer. Despite ambiguous results on the concentration of FKN in various breast cancer tissue samples, in vitro studies demonstrated the influence of FKN on the migration and invasion abilities of cancer cells mediated via the Src family kinase (Src)/focal adhesion kinase (FAK) signaling pathway following EGFR activation by ADAMs. Given the relatively high expression of FKN in spinal cancellous bone, CX3CR1-expressing metastatic tumor cells may be attracted [338]. An in vitro study demonstrated that the Src/FAK signaling pathway played a vital role in the FKN-dependent promotion of lung cancer cell migration and invasion [339].

#### 5.2.2. FKN/CX3CR1 Signaling May Be a Good Prognostic Factor in Cancer

However, publications indicating the adverse effects of the FKN/CX3CR1 axis in promoting the growth and spread of various cancers are accompanied by an increasing number of contradictory reports. These cancers include frequently diagnosed cancers, such as colorectal cancer, breast cancer, and lung cancer, in which the absence or low expression of FKN/CX3CR1 in tumor tissue is a poor prognostic factor associated with an increased risk of metastatic progression. The explanation for these observations may be clearly related to the activity of the immune response toward the tumor, which is determined by the chemotaxis of CX3XR1+ cells [340].

For example, the CX3CR1 gene has been identified as a hub gene in colorectal cancer, i.e., a gene that interacts with many other genes in the gene network and commonly plays a critical role in biological processes and gene regulation in the course of the disease [341,342,343]. Using the Tumor IMmune Estimation Resource (TIMER) database and CIBERSORT analysis, correlations between CX3CR1 and tumor-infiltrating immune cells were estimated in Yue et al. [344]. A co-culture of the human monocytic cell line THP-1-derived macrophages with the human colon carcinoma cell line HCT8 with low CX3CR1 expression was established in which immune marker expression, cell viability, and migration were investigated. Patients with low immune marker expression scores had significantly shorter survival than patients with high immune marker expression scores. CX3CR1 may act as a prognostic biomarker in colorectal cancer because its expression is associated with immune marker expression, immune cell infiltration levels, and macrophage polarization. CX3CR1 expression determines the recruitment and regulation of immune-infiltrating cells and macrophage polarization in colorectal cancer. Therefore, the silencing of CX3CR1 may promote the proliferation and migration of colorectal cancer cells [344].

Another study showed that the co-expression of FKN and CX3CR1 in colorectal cancer cells (FKN-CX3CR1 axis-positive tumors) was associated with a significantly longer period of disease-free and disease-specific survival [345]. Therefore, the appropriate level of FKN-CX3CR1 axis activity in tumor cells acts as a retention factor, which likely increases homotypic cell adhesion and limits tumor spreading to metastatic sites. Conversely, no or low expression of FKN-CX3CR1 in axis-negative colorectal cancer cells poses an increased risk of tumor relapse and an increased likelihood of metachronous metastasis [345].

FKN overexpression may be a predictive biomarker for identifying antibody-dependent cellular cytotoxicity (ADCC)-based therapy responders [346]. FKN overexpression attracted tumor-suppressive lymphocytes, including NK cells, and inhibited tumor growth and lung metastasis in a syngeneic 4T1 cell line in a mouse model of breast cancer. Increased NK cell-mediated cytotoxicity acted synergistically with trastuzumab, a humanized anti-HER2 oncogene monoclonal antibody, for the treatment of breast cancer [346]. FKN overexpression in humanized tumor mice, which show human tumors and the human immune system, resulted in the enhanced efficacy of trastuzumab treatment, especially for preventing lung metastases composed of FKN-overexpressing breast cancer cells [347].

The tumor-growth-inhibiting effect of FKN, which is a derivative of increased NK cell activity, was also observed in an orthotopic implantation of a lung cancer model in vivo [348]. Analysis of lung cancer data from the Gene Expression Omnibus database and The Cancer Genome Atlas revealed that increased FKN mRNA expression in tumor tissues from lung adenocarcinoma patients was associated with improved overall survival (OS) and, thus, a positive prognostic factor [349].

Similarly, the prognostically favorable effect of the FKN-CX3CR1 axis, involving the recruitment of cytotoxic T cells, NK cells, and DCs to the TME, was demonstrated in hepatocellular carcinoma and gastric adenocarcinoma [350,351].

#### 5.2.3. Possible Reasons for the Contradictory Results of FKN/CX3CR1 Signaling in Cancer

As cited in Section 5.2.1 and Section 5.2.2, the results on the role of the FKN/CX3CR1 axis in common cancers remain clearly contradictory [88]. Although we notice the fragmentary nature of these reports, these results indicate that signaling involving this peculiar chemokine with adhesion molecule properties is more complex [352].

In addition to the heterogeneity of tumors mentioned in this chapter, the explanation for this surprising discrepancy in the interpretation of the impact of FKN (favorable vs. unfavorable) on the course of cancer may be that the pro-inflammatory, anti-apoptotic, and pro-angiogenic effects mediated by the FKN/CX3CR1 axis are in opposition to the impact of this signaling pathway on immune system functions. CX3CR1-positive leukocytes, including CD4+ and CD8+ T cells, monocytes, B cells, neutrophils, natural killer (NK) cells, and dendritic cells (DCs), are subject to Sfkn-related chemotaxis [353]. Therefore, FKN is an important tumor-infiltrating lymphocyte (TIL)-recruiting chemokine and a key regulator of cytotoxic T-cell-mediated immunity. There is also growing evidence that the FKN pathway is involved in the maintenance of effector memory cytotoxic T-cell populations responsible for anti-viral and anti-tumor immunity [88].

The explanation for the efficacy of the immune response related to the FKN/CX3CR1 axis in various types of cancers (anti-tumor response) or lead to disease progression (pro-tumor response) is related to the presence of two forms of FKN: membrane-bound and soluble FKN. The upregulated expression of FKN leads to the increased accumulation of CX3CR1+ immune system cells in tumor tissue and significantly increases local CX3CR1 density with the possibility of CX3CR1 induction in tumor cells [150,344]. The simultaneous co-expression of mFKN and CX3CR1 in cancer cells leads to cell adhesion, which significantly impedes cellular migration and tumor spread [345].

However, this beneficial effect of FKN/CX3CR1 signaling may not occur when mFKN is cleaved by proteinases, such as ADAM10 or TACE/ADAM17, and the mFKN/sFKN balance is shifted in favor of the soluble form. Under these conditions, tumor cells no longer adhere to each other or adhere much weaker because the dominant CX3CR1 ligand becomes sFKN, which promotes chemotaxis. An increase in CX3CR1 expression increases sFKN-induced cancer cell migration [354]. The source of sFKN may be the cancer cells themselves and other TME components, e.g., fibroblasts [355,356]. Therefore, simultaneous increases in sFKN and CX3CR1 expression may be responsible for the pro-tumor effect of the FKN/CX3CR1 axis, which increases the risk of metastasis [332].

The transient or tumor-specific activity of ADAM10 or TACE/ADAM17 may play a key role because changing the mFKN/sFKN ratio modifies the signaling of the FKN/CX3CR1 axis to reveal anti- or pro-tumor effects [357,358].

Overall, the uncontrolled proliferation and apoptosis resistance of cancer cells do not necessarily depend directly on CX3CR1 but rather on EFGR signaling [102]. This effect may be difficult to separate because some downstream signaling pathways related to FKN, including the PI3K/Akt/IKK/Iκβ/NF-κβ pathways, are also activated by EGFR stimulation [180,189]. Moreover, in the case of signaling through EGFR, the signaling pathways are at least as complex, as can be demonstrated by the MAPK signaling network [359]. Therefore, further detailed research into the nature of these very complex phenomena is necessary, the analysis of which should take into account tools created using artificial intelligence (AI), in particular specific machine learning (ML) paradigms [360,361,362].

The mechanisms determining the pro-tumor or anti-tumor effects of FKN/CX3CR1 signaling in cancer are summarized in Figure 5.

## 6. Concluding Remarks

The FKN/CX3CR1 signaling pathway plays an important role in angiogenesis, especially in the chronic inflammatory response. This role also applies to angiogenesis in the TME, where typical stimuli are hypoxia and inflammation, which modulate cell polarization and induce cell plasticity to promote tumorigenesis. Beneficial and detrimental effects are observed in angiogenesis accompanying inflammation, and angiogenesis within the TME is always related to tumor growth and metastasis. Therefore, there is an important need to determine how the FKN-CX3CR1 axis influences the course of cancer for therapeutic reasons. The difficulty is that the effect of FKN on the functioning of the immune system in the TME is complex and depends on the type of cancer and the heterogeneity of a specific tumor, which affects the expression of FKN and CX3CR1 in cells. The variable proportions of two forms of FKN, membrane-bound (mFKN) and soluble (sFKN), are responsible for adhesion and chemotaxis, respectively, and may determine the occurrence of anti-tumor or pro-tumor effects. Therefore, FKN/CX3CR1 may be a favorable or unfavorable prognostic factor, with evidence for anti-tumor functions primarily coming from prognostic studies, while evidence for pro-tumor function tends to be from tumor development studies. This paradoxical situation suggests that the FKN-CX3CR1 axis is a double-edged sword in cancer biology, which reveals a conflict between therapeutic goals. The essence of the development of new treatment methods must be to increase the FKN-dependent recruitment of CX3CR1+ immune cells, such as NK cells, DCs, and CD4+ and CD8+ T cells, to the TME. The homing of these cells to the TME may be enhanced by increasing CX3CR1 expression and/or increasing sFKN expression in the tumor. However, the increased expression of CX3CR1 in cancer cells that do not express mFKN and the migration of these cells under the influence of the chemotactic effect of sFKN should be prevented because it promotes tumor growth via angiogenesis and increases the risk of metastasis. Limiting treatment only to the FKN/CX3CR1 signaling pathway will not be fully effective due to the numerous processes involved in tumors, which result in cancer immune evasion.

## Figures and Tables

**Figure 1 ijms-25-04679-f001:**
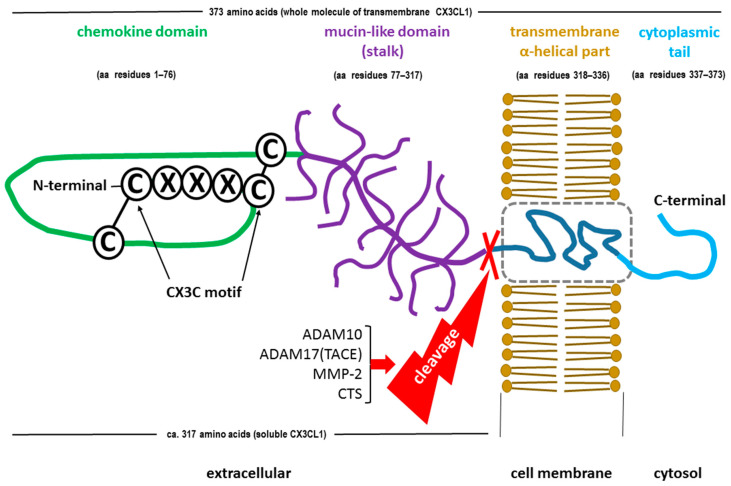
Schematic structure of the fractalkine (FKN, chemokine CX3CL1) molecule as a transmembrane protein, including the soluble form (sFKN) cleaved by a disintegrin and metalloproteinase domain-containing protein 10 (ADAM10), tumor necrosis factor alpha (TNF-α) converting enzyme (TACE or ADAM17), matrix metalloproteinase-2 (MMP-2), or cathepsins (CTS).

**Figure 2 ijms-25-04679-f002:**
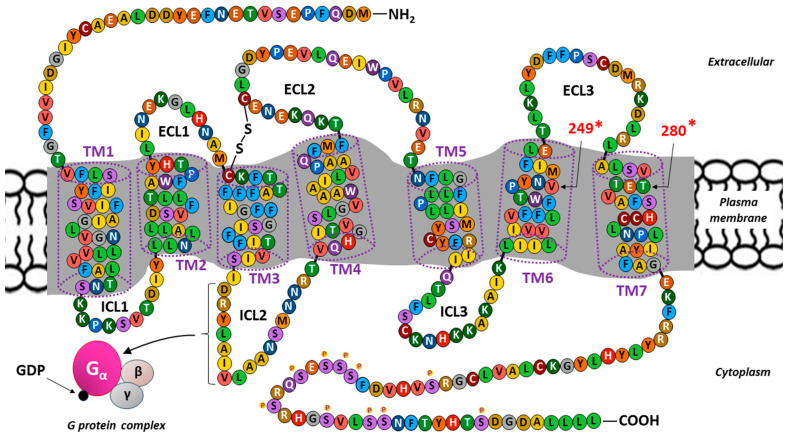
Schematic diagram and amino acid sequence of CX3C motif chemokine receptor 1 (CX3CR1), the only fractalkine (FKN, chemokine CX3CL1) receptor. Belonging to the most numerous class A (rhodopsin-like receptors) in the G protein-coupled receptor (GPCR) family of proteins, CX3CR1 shares a common structural signature with the polypeptide chain (355 aa; ∼40 kDa), containing seven hydrophobic α-helical transmembrane (TM1–TM7) segments or domains, with an extracellular amino terminus (NH2) and an intracellular carboxyl terminus (COOH). These transmembrane segments are connected to each other by three intracellular (ICL1–ICL3) and three extracellular loops (ECL1–ECL3) [107,108]. A disulfide bridge is marked, connecting two conservative cysteine (C) residues located at the top of the extracellular side of TM3 and within ECL2, respectively [109]. The ICL2 contains the canonical DRYLAIV motif, composed of a sequence of seven amino acids (in 3-letter abbreviations: Asp-Arg-Tyr-Leu-Ala-Ile-Val) forming a docking site that is essential for CX3CR1 coupling to the G protein and the induction of classical signaling pathways [104,110]. The binding of an agonist to CX3CR1 causes conformational changes in the receptor with the subsequent dissociation of the components of the heterotrimeric G complex, consisting of alpha (α), beta (β), and gamma (γ) subunits. Binding of guanosine diphosphate (GDP) enables the α subunit to bind to the β and γ subunits to form an inactive trimer. The binding of an extracellular signal (ligand) to CX3CR1 allows the G protein to bind to the receptor and causes GDP to be substituted by guanosine triphosphate (GTP; not shown) [106,111]. The COOH-terminal serine residues (S) are susceptible to G protein-coupled receptor kinase (GRK)-mediated phosphorylation (marked with tiny yellowish dots containing P). * The polymorphic residues at positions 249 and 280 may be responsible for dysfunctional CX3CR1 variants [112,113].

**Figure 3 ijms-25-04679-f003:**
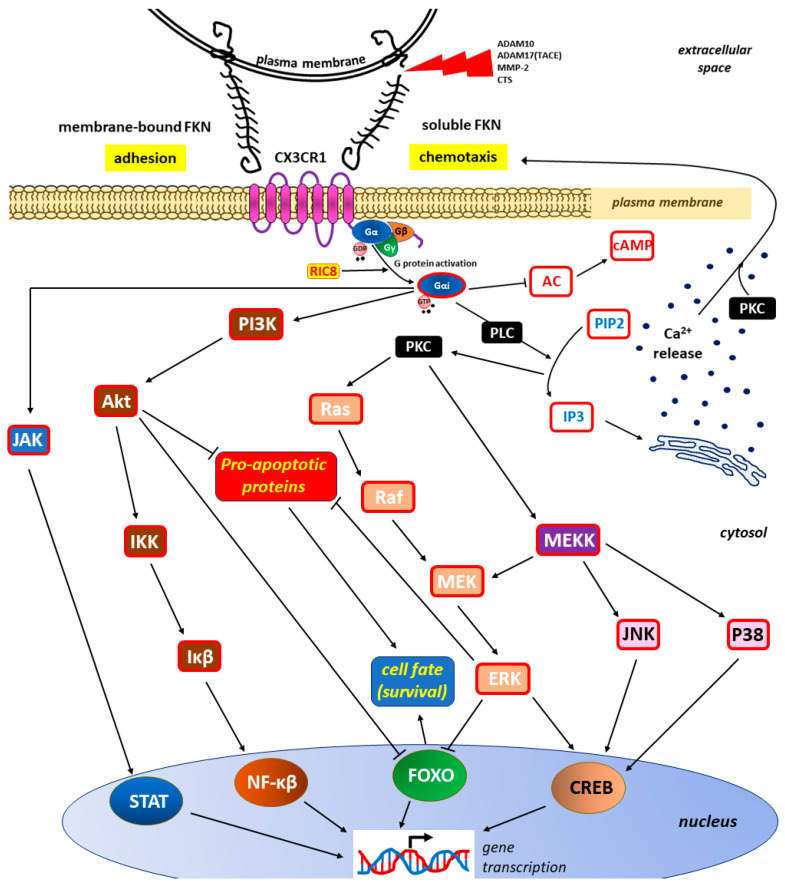
Main signaling pathways initiated via the activation of the CX3C motif chemokine receptor 1 (CX3CR1) receptor through the binding of its sole endogenous ligand FKN (fractalkine, chemokine CX3CL1). For the sake of clarity, interactions with other receptors have been omitted.

**Figure 4 ijms-25-04679-f004:**
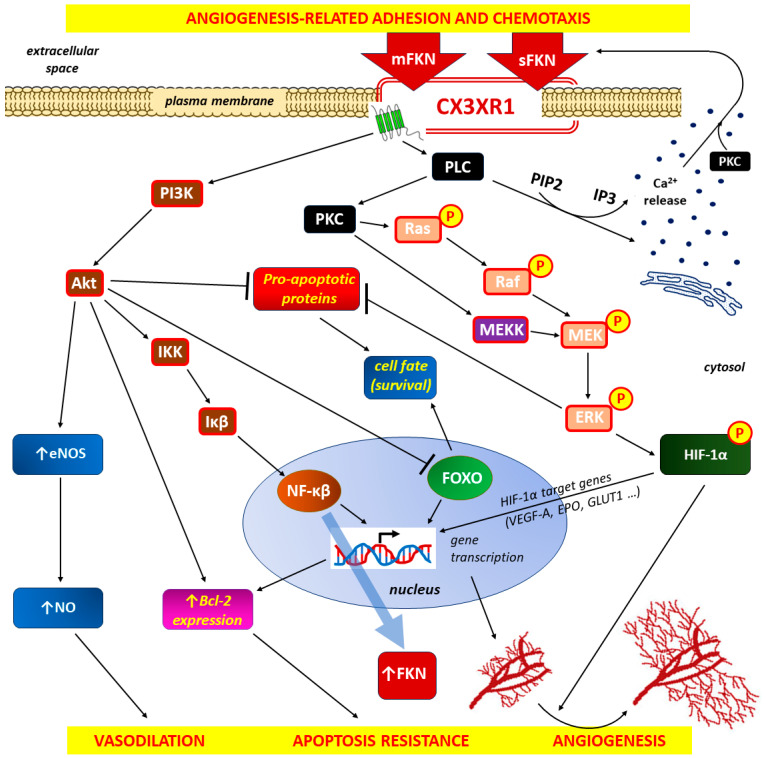
Angiogenesis during (chronic) inflammatory response as a consequence of specific signaling through the FKN/CX3CR1 axis (for the full spectrum of signaling via CX3CR1, see Figure 3).

**Figure 5 ijms-25-04679-f005:**
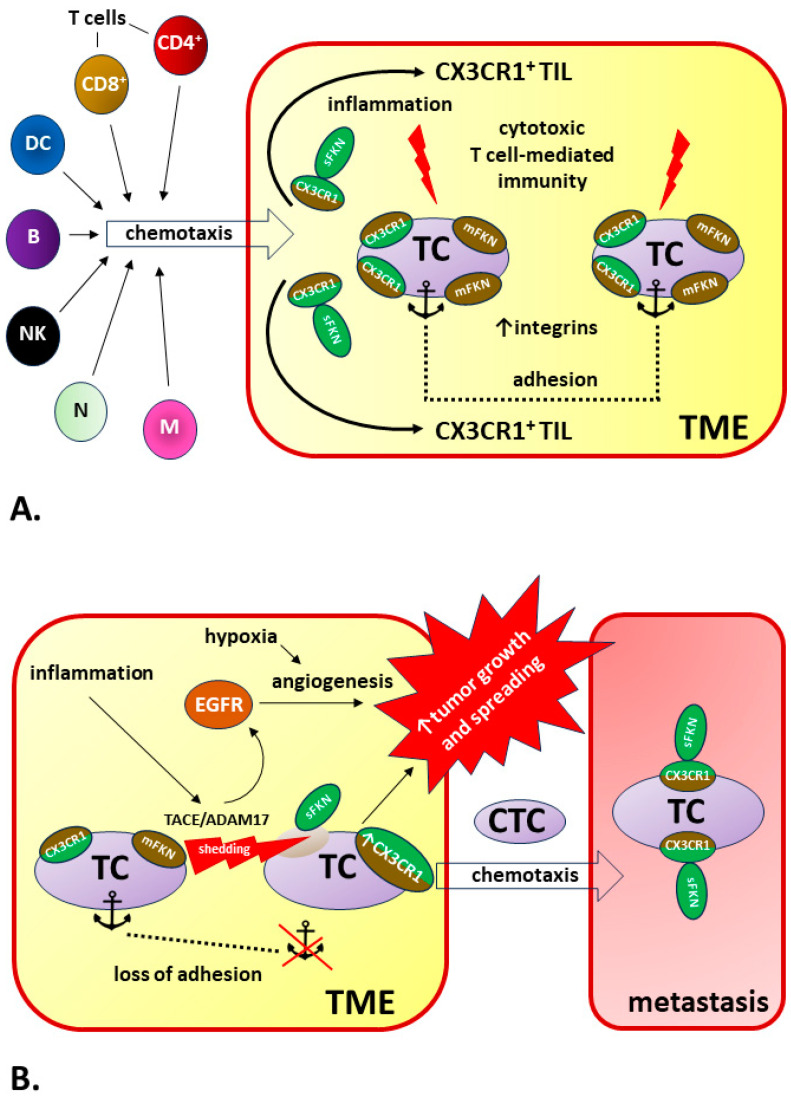
The opposing actions of the FKN/CX3CR1 axis in the tumor microenvironment (TME) explain the occurrence of both anti-tumor (**A**) and pro-tumor (**B**) effects. (**A**) Chronic inflammation in the tumor microenvironment (TME) upregulates both forms of fractalkine (FKN). The soluble form (sFKN) acts as a chemoattractant bringing cytotoxic and cytokine-producing cells to areas of inflammation. Thus, FKN serves as an important chemokine recruiting CX3CR1+ tumor-infiltrating leukocytes (TIL) such as CD4+ and CD8+ T cells, monocytes (M), B cells (**B**), neutrophils (N), natural killer (NK), and dendritic cells (DC) [353]. Enhanced cytotoxic T-cell-mediated immunity is associated with **anti-tumor effects**. Additionally, increased CX3CR1 density in the TME, resulting from the accumulation of CX3CR1+ cells of the immune system, may induce CX3CR1 expression in tumor cells (TC) [150,344]. Simultaneous co-expression of the membrane-bound FKN (mFKN) and CX3CR1 in cancer cells activates integrins, which leads to cells adhering and sticking together (marked with anchors), which significantly impedes their migration and the spread of the tumor [345]. (**B**) Chronic inflammation in TME may activate proteinases, mainly tumor necrosis factor-alpha converting enzyme (TACE), also known as a disintegrin and metalloprotease 17 (ADAM17), TACE/ADAM17, transiently or in a tumor-specific manner [357,358]. As a result, mFKN on TC is cleaved, and the mFKN/sFKN balance is shifted in favor of the soluble form. TC adhesion is significantly weakened (marked with a crossed out anchor) because sFKN, responsible for chemotaxis, has become the main CX3CR1 ligand. If this is accompanied by the increased expression of CX3CR1 in TC, sFKN-induced cancer cell migration will occur [354]. Therefore, simultaneous increases in sFKN and CX3CR1 expression may have **pro-tumor effects** on the FKN/CX3CR1 axis with an increased risk of metastasis due to the presence of circulating tumor cells (CTCs), separated from the primary tumor and traveling through the blood stream [332]. Because TACE/ADAM17 activates epidermal growth factor receptor (EGFR), hypoxia-induced angiogenesis, uncontrolled proliferation, and the apoptosis resistance of cancer cells do not necessarily depend directly on CX3CR1 but on EFGR signaling [98]. Moreover, some FKN signaling pathways, including PI3K/Akt/IKK/Iκβ/NF-κβ, are also EGFR-mediated [180,189].

**Table 1 ijms-25-04679-t001:** Examples of proven actions within the FKN/CX3CR1 axis in physiological states and the pathophysiology of diseases in selected tissues, organs, and systems.

Location in the Body	Regulation of Biological Processes via FKN/CX3CR1 Axis in Normal and Pathological Conditions	References
Central nervous system (CNS)	- In brain tissues, FKN is mostly expressed in neurons, while microglia express CX3CR1. - FKN/CX3CR1 signaling enables precise interactions between neurons, microglia, and immune cells. - Due to its key role in microglia–neuron communication, the FKN/CX3CR1 axis regulates a broad spectrum of microglial properties, including microglial cell migration and dynamic surveillance of the brain parenchyma, neuronal integrity and survival, synaptic plasticity, and diverse synaptic functions, as well as neuronal sensitivity to stimuli and excitability via cytokine release modulation, chemotaxis, and phagocytosis. - FKN suppresses lipopolysaccharide (LPS)-induced microglia activation by reducing the production of nitric oxide (NO), interleukin-6 (IL-6), and transforming growth factor alpha (TNF-α), therefore inhibiting neuronal cell death in response to LPS cytotoxicity in the brain tissue. - FKN/CX3CR1 signal disruption is is one of the most important phenomena in the pathomechanisms of CNS-related disorders, especially neurodegenerative diseases and traumatic brain injuries. However, the results of studies on the modulation of inflammation in the CNS by FKN/CX3CR1 are often ambiguous or contradictory. For example, the disruption of FKN signaling is beneficial in limiting the effects of CNS ischemia but detrimental in other neurodegenerative diseases, including Parkinson’s disease (PD) and amyotrophic lateral sclerosis (ALS). Furthermore, the deletion of CX3CR1 in Alzheimer’s disease may, possibly depending on the stage of disease progression, lead to both neuroprotective and detrimental effects. There is also no complete agreement on the importance of the involvement of FKN isoforms in the development of neuropathological processes. - The type of specific response, neurotoxic or neuroprotective, most often depends on the type of damaging factor, the CNS area influencing the regional heterogeneity of microglial cells, and the local concentrations of FKN and CX3CR1.	Sheridan and Murphy 2013 [192] Pawelec et al., 2020 [193] Paolicelli et al., 2014 [194]; Camacho-Hernández and Peña-Ortega 2023 [195] Mizuno et al., 2003 [196]; Lyons et al., 2009 [165]; Mecca et al., 2018 [197] Subbarayan et al., 2022 [198]; Cipriani et al., 2011 [199]; Poniatowski et al., 2017 [61]; Luo et al., 2019 [200]; Bivona et al., 2023 [201]; Nash et al., 2015 [202]; Juliani et al., 2021 [203]; Lee et al., 2010 [204], Cho et al., 2011 [205]; Fuhrmann et al., 2010 [206]; Pawelec et al., 2020 [193]; Eugenín et al., 2023 [207] Sheridan and Murphy 2013 [192]; Stratoulias et al. [208]
Bone marrow and immune tissue	- The expression of CX3XR1 increases with the maturation of myeloid cells and shows an inverse correlation with the Ly6C marker and the C-C chemokine receptor 2 (CCR2) in the blood. This may indicate that CX3CR1 limits the motility of Ly6C (high) monocytes within the bone marrow and, thus, controls their release into the blood. - FKN-CX3CL1 axis plays a significant role in an early stage of osteoblast differentiation, possibly through their trans and cis interactions. - FKN regulates mouse osteoclast precursor (OCP) survival and primes OCPs for subsequent osteoclast differentiation. - Autoimmune and inflammatory responses in rheumatoid arthritis (RA) are positively correlated with the concentration of FKN in the serum and synovial fluid. The associated chemotaxis primarily involves the recruitment of CD16+ monocytes into synovial tissues, as CX3CR1 expression in CD16+ monocytes is markedly higher compared to other populations (e.g., CD14+ and CD167−). - Bone marrow (BM) FKN levels are significantly increased in the multiple myeloma (MM) patients and positively correlated with BM microvessel density. - CX3CR1 expression is an additional marker of natural killer (NK) cell differentiation and closely related to their ability to migrate to the central nervous system (CNS) from the periphery. - CX3CR1 is prevalently expressed on killer cell lectin-like receptor subfamily G member 1 (KLRG1)^+^ NK cells, a subset that is considered terminally differentiated. Therefore, CX3CR1 may represent a marker of a KLRG1(+) NK-cell subset with its own unique properties that can unidirectionally and irreversibly differentiate from the KLRG1(+)/CX3CR1(−) NK cells during a functionally stable period of stay in the bone marrow. - FKN activates the Jak2-Stat5α-ERK1/2 pathway via CX3CR1, thereby triggering integrin-dependent mechanisms of cytoskeleton remodeling to allow chemotactic relocations of bone marrow-derived mesenchymal stem cells (BMSCs) toward an ischemic cerebral lesion.	Jacquelin et al., 2013 [209] Hoshino et al., 2013 [210] Kuboi et al., 2022 [211] Yano et al., 2007 [212] Marchica et al., 2019 [213] Hamman et al., 2011 [94] Sciumè et al., 2011 [214] Zhang et al., 2015 [215]
Cardiovascular system	- FKN and CX3CR1 are expressed in atherosclerotic lesions, and FKN is involved in the initiation step of atherosclerotic plaque formation. Monocyte–endothelial cell interactions are partly mediated by the expression of the monocyte CX3CR1 and endothelial cell FKN. The activation of these lymphocytes upon ligand/receptor binding leads to the release of lysis granules that destroy the vascular endothelium. - After endothelial damage, the release of FKN from apoptotic cells results in subsequent recruitment of macrophages, which promotes the removal of apoptotic debris; however, in more advanced stages of atherosclerosis, signaling through the FKN/CX3CR1 axis enhances foam cell formation, promoting the development of atherosclerotic plaques. - CX3CR1 expression on vascular smooth muscle cells (VSMCs) within atherosclerotic plaque causes the functional state of the FKN/CX3CR1 axis to play an important role in plaque stability. Emergency conditions associated with cardiovascular mortality and morbidity are typically caused by the rupture of “vulnerable” atherosclerotic lesions. - FKN promotes the development of atherosclerotic lesions by activating platelets and causing their adhesion to the endothelium. - The early activation of the cardiac FKN/CX3CR1 axis delays and limits β-adrenergic-induced heart failure. - FKN levels are markedly elevated during acute myocardial infarction (AMI) compared to patients with stable angina pectoris (AP), although they do not correlate with infarct size. The inverse pattern in gene expression of CX3CR1 might be here a kind of compensatory mechanism. - In addition to demonstrating a positive correlation of FKN concentration with an increased risk of developing poorer cardiac function after AMI, levels of FKN can also be treated as a prognostic for the risk of developing major adverse cardiovascular events (MACEs) in acute ST-elevation myocardial infarction (STEMI) patients. - The inhibition of the FKN/CX3CR1 interaction has a beneficial effect on the final infarct size after reperfusion, as it reduces the severity of an important complication—ischemia/reperfusion injury. This complication is directly related to the action of CX3CR1^+^ lymphocytes toward microvascular obstruction (MVO). - Increased risk of deep vein thrombosis (DVT) is positively correlated with increased activity of the FKN/CX3CR1 axis that involves CX3CR1-expressing platelets, then binding to monocytes and CD8+ lymphocytes. - In metabolic syndrome, platelet activation occurs and the percentage of platelet–eosinophil and platelet–lymphocyte aggregates increases, which is accompanied by the upregulation of platelet CX3CR1 expression. FKN-dependent increased adhesion of these aggregates may play a key role in atherogenesis.	Teupser et al., 2004 [216]; Ma et al., 2022 [217]; Riopel et al., 2019 [218]; Liu and Jiang 2011 [219] Elliott et al., 2017 [220]; White et al., 2014 [190]; Landsman et al., 2009 [171] Lucas et al., 2003 [221]; Harman and Jørgensen 2019 [222]; Apostolakis and Spandidos 2013 [223]; Skoda et al., 2018 [224] Noels et al., 2019 [225]; Flierl et al., 2015 [80] Flamant et al., 2021 [226] Njerve et al., 2014 [227]; Yao et al., 2015 [228] Yao et al., 2015 [228]; Xu et al., 2019 [229] Loh et al., 2023 [44]; Boag et al., 2015 [230] Furio et al., 2018 [231] Marques et al., 2019 [232]
Respiratory system	- CX3CR1^+^ leukocyte attachment to the lung vascular endothelium and diapedesis through the glycocalyx, endothelial cell layer and the basement membrane lead to mononuclear cell accumulation in the lung vessel walls and parenchyma. Infiltrated CX3CR1+ immune cells are a source of mediators that induce damage, stimulate proliferation, and/or affect chemoattract inflammatory cells. The result of these cumulative actions is a structural destruction and remodeling in the development of inflammatory lung diseases. - FKN/CX3CR1 signaling may be involved in the pathophysiology of hypoxia-induced pulmonary arterial hypertension (PAH) developing due to chronic inflammation. Both increased FKN concentrations and upregulated CX3CR1 expression cause PAH progression with vascular remodeling and proliferation of pulmonary artery smooth muscle cells. - Soluble FKN chemoattracts and activates CX3CR1+ leukocytes such as CD8+, CD4+, and γδ T lymphocytes; natural killer cells; dendritic cells; and monocytes/macrophages, leading to mononuclear cell influx and accumulation in the lung vessel walls and parenchyma. During the resolution phase of acute lung injury, apoptotic cell-derived CX3CL1 attracts alveolar macrophages transmigration toward apoptotic cells for phagocytosis. - In allergic asthma, CX3CR1 signaling is essential for airway inflammation by promoting T helper cell survival and maintenance in inflamed lung together with chemotaxis recruited mast cells into bronchial mucosa. - FKN is elevated in both bronchoalveolar lavage fluid and sputum from human asthmatics sensitized to fungi, implicating an association with FKN in fungal asthma severity. However, FKN/CX3CR1 axis preserves lung function during fungal-associated allergic airway inflammation through a nonclassical immunoregulatory mechanism. Hence, the knockout of CX3CR1 signaling resulted in a profound impairment in lung function during fungal-associated allergic airway inflammation. - In pulmonary infections, the role of FKN/CX3CR1 axis remains unclear. For example, FKN may be involved in both immunopathological and anti-viral immune responses to rhinovirus infection.	Zhang and Patel 2010 [233] Balabanian et al., 2002 [234]; Amsellem et al., 2017 [235] Tsai et al., 2021 [236] Mionnet et al., 2010 [95]; El-Shazly et al., 2006 [237] Godwin et al., 2021 [238] Upton et al., 2017 [239]
Liver	- FKN/CX3CR1 is upregulated during liver damage including chronic inflammatory liver diseases such as chronic hepatitis C, nonalcoholic steatohepatitis (NASH)/nonalcoholic fatty liver disease (NAFLD), and cirrhosis. - The assessment of the impact of increased FKN/CX3CR1 activity on the severity of steatosis, inflammation, and liver fibrosis is still ambiguous. In addition to reports indicating that FKN-CX3CR1 interaction limits inflammatory properties in Kupffer cells/macrophages, resulting in a reduction in liver inflammation intensity and decreased fibrosis, there are also contradictory research data. - FKN/CX3CR1 upregulation was reported in injured bile ducts of primary cirrhosis with its involvement in the recruitment of intraepithelial lymphocytes of intrahepatic bile ducts. Moreover, the correlation between primary biliary cirrhosis and FKN expression is significantly proportional.	Efsen et al., 2002 [240]; Bourd-Boittin et al., 2009 [63]; Sutti et al., 2015 [241]; Nagata et al., 2022 [242] Aoyama et al., 2010 [243]; Zhang et al., 2020 [244]; Ni et al., 2022 [75]; Sutti et al., 2015 [241]; Karlmark et al., 2010 [245]; Wasmuth et al., 2008 [246]; Hassan et al., 2023 [247] Isse et al., 2005 [248]; Shimoda et al., 2010 [249]
Gut	- Most macrophages and some dendritic cell (DC) subsets express CX3CR1 in the gut. In resting intestinal mucosa, the role of lamina propria CX3CR1^+^ macrophage is to pass captured antigen via trans-epithelial dendrites or phagocytosis onto DC for transport to the mesenteric lymph node (MLN) to prime immune responses like lamina propria DC. - The deletion of FKN or CX3CR1 leads to a specific and significant reduction in lamina propria macrophages with reductions in the translocation of bacteria to MLNs and their ability to take up pathogens. Therefore, CX3CR1 may be treated as a specific marker useful for lamina propria macrophages and a key component in sustainment lamina propria macrophage homeostasis. Contradictory, it was demonstrated that CX3CR1 knockout mice show normal numbers of macrophages. - The intestinal microbiome influences the local accumulation of CX3CR1^+^ phagocytes, and the number of CX3CR1^+^ cells is reduced in germ-free mouse. - The enhanced recruitment of CX3CR1^+^ T cells by mucosal human intestinal microvascular endothelial cell (HIMECs)-derived FKN has been demonstrated in inflammatory bowel disease (IBD).	Joeris et al., 2017 [250]; Niess et al., 2005 [118]; Bain and Mowat 2011 [251]; Lee et al., 2018 [74] Ferretti et al., 2014 [252]; Bain et al., 2013 [253] Bain et al., 2014 [254] Sans et al., 2007 [255]
Placenta	- Human placenta is a source of FKN, which is expressed in the syncytiotrophoblast and can be released into the maternal vascular compartment (maternal circulation) by constitutive MMP-dependent shedding. - FKN content within the apical microvillous plasma membrane increases significantly in the placenta of full-term pregnancy compared to the first trimester. - FKN/CX3CR1 axis mediates the adhesion of monocytes to the villous trophoblast. - Increased expression and release of placental FKN may be responsible for low-grade systemic inflammatory background and responses in the third trimester of a normal pregnancy. - Placental FKN is upregulated in severe early-onset pre-eclampsia (PE). Significant underdevelopment of placental vascular network with a significantly lowered vascular/extravascular tissue index (V/EVTI) in PE is associated with the dysregulation of the FKN/CX3CR1 system, especially in fetal growth restriction (FGR)-complicated pregnancies. - Increased average FKN content in the diabetic placenta is accompanied by an increase in the density of placental microvessels and a higher expression of CX3CR1 compared to the placenta from a normal pregnancy. Therefore, FKN/CX3CR1 signaling pathway is involved in the pathomechanism of placental microvasculature remodeling during diabetes class C (after White). - Placental hypoxia increases FKN production and upregulates CX3CR1 expression in the placental endothelial cells. Under these conditions, tumor necrosis factor alpha (TNFα) induces FKN, influencing a mechanism of FKN autoregulation via CX3CR1 expression. - Increased FKN concentration, accompanied by a higher mean FKN gene expression level in the tissues of pregnant women with missed abortion, may be responsible for abnormal placental invasion.	Siwetz et al., 2014 [256] Siwetz et al., 2015 [257] Siwetz et al., 2015 [258] Vishnyakova et al., 2021 [259] Szewczyk et al., 2021 [260]; Ullah et al., 2023 [261] Szukiewicz et al., 2013 [262]; Ullah et al., 2023 [261]; Szukiewicz et al., 2014 [263] Gokce et al., 2022 [264]
Joint and bone tissue	- The total number of circulating CX3CR1^high^ T cells is increased in the circulation of rheumatoid arthritis (RA) patients. Joint-infiltrated CX3CR1^high^ T cells tightly and strongly adhere to fibroblast-like synoviocytes (FLSs) in the synovium in an FKN-dependent manner. - The FKN/CX3CR1 axis promotes inflammation-free osteoclastogenesis by enhancing precursor cell survival and differentiation. - The apoptosis of chondrocytes during joint osteoarthritis upregulates the FKN-CX3CR1-p38 axis, which results in the enhanced chemotaxis of osteoclast precursors (OCPs) and promotes bone resorption. - The development of osteoarthritis (OA) is largely driven by low-grade local background inflammation based on FKN-mediated chemotaxis. - FKN/CX3CR1 signaling in hemophilia is involved in the pathomechanism of irreversible joint degeneration (hemophilic arthropathy). - Significantly increased concentrations of FKN in human blood serum are accompanied by high concentrations of bone turnover and inflammatory factors in the serum, such as tartrate-resistant acid phosphatase 5b (TRACP-5b), cross-linked N-telopeptides of type I collagen (NTx), and interleukins (IL-1β, IL-6). - FKN knockdown ameliorates inflammation and apoptosis after exposure to LPS and accelerates osteogenic differentiation. These effects related to FKN deficiency can be reversed by increased expression of CX3CR1. - FKN axis signaling alleviates intervertebral disc degeneration (IDD) by reducing inflammation and apoptosis of human nucleus pulposus cells (HNPCs) via macrophages.	Tanaka et al., 2020 [265] Kuboi et al., 2022 [211]; Koizumi et al., 2009 [266] Koizumi et al., 2009 [266]; Guo et al., 2022 [267] Wojdasiewicz et al., 2014 [43,268] Wojdasiewicz et al., 2020 [188] Wojdasiewicz et al., 2019 [164] Lu et al., 2023 [269] Gao et al., 2023 [270]

## Data Availability

No new data were created. Instead, the data are quoted from the available cited literature.

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
