# Peer review of "CX3CL1 (Fractalkine)-CX3CR1 Axis in Inflammation-Induced Angiogenesis and Tumorigenesis"

_ijms, 2024, doi:10.3390/ijms25094679_

Round 1
Reviewer 1 Report
Comments and Suggestions for Authors
This manuscript (review) addresses the importance of CX3CL1 (Fractalkine; FKN) and its receptor CX3CR1 for signalling pathways in tumorigenesis and cancer metastasis based on the effects on cell adhesion, apoptosis and cell migration. The author aims to focus on the role of the FKN signalling pathway in the context of angiogenesis in inflammation and cancer. One important goal was to present mechanisms determining the pro- or anti-tumor effects, which are often contradictory and create confusion in the field.
This is a very nice, timely and important review on CX3CL1 (FKN)/ CX3CR1 and inflammation, written by an expert in this field. This referee has only some points which the author may want to consider.
1) Cite other reviews? This manuscript focuses on the FKN signalling pathway and its role for angiogenesis in inflammation and cancer. However, as mentioned, there are at least 45 members of human chemokines and their corresponding receptors. It will be helpful for the future readership if the authors could highlight a very limited number of other recent reviews which summarize important aspect of chemokine biology/pathophysiology not covered here.
2) Cell-type specific effects. In the manuscript, often signalling pathways are discussed in general, not cell-specific terms. For example, are figures 1-5 representative for most cells discussed in the paper, or only for some selected ones? Perhaps, Table 1 is meant to show this and can be made more readable. Also, what about human platelets? They have important chemokines /receptors with functions.
3) Bi-(multi)-directional regulation. The discussion on the mechanisms determining the pro- or anti-tumor effects in this system is very important and underscores the complexity of this type of regulation (also Fig. 5). These are increasingly observed with complex signalling pathways such as MAPKs and others. [Peterson AF et al Systematic analysis of the MAPK signaling network... Cell Systems 2022; 13: 885-894.e884. DOI: 10.1016/j.cels.2022.10.003 ]
4) Conclusions. Is there any perspective to elucidate such complex regulatory patterns?
Comments on the Quality of English LanguageFine.
Author Response
"Please see the attachment"

Reviewer 2 Report
Comments and Suggestions for Authors
The manuscript entitled “CX3CL1 (Fractalkine)-CX3CR1 axis in inflammation-induced angiogenesis and tumorigenesis” is a very long literature review article describing the role of CX3CL1 in different processes including angiogenesis and tumorigenesis. Angiogenesis and tumorigenesis are interconnected topics and authors should summarize common aspects to decrease the article length. The manuscript presents 39% of a match with previously published papers. Therefore, modify the sentences to decrease similarity. Is too high for a review paper. I suggest modification in the review structure. The author should write an introduction section introducing the review topic and after that, show subheadings with paper topics. Therefore, the manuscript is interesting and should be reorganized to increase readability.
Author Response
"Please see the attachment".
